# Aptamer-Based Proteomics Measuring Preoperative Cerebrospinal Fluid Protein Alterations Associated with Postoperative Delirium

**DOI:** 10.3390/biom13091395

**Published:** 2023-09-15

**Authors:** Simon T. Dillon, Sarinnapha M. Vasunilashorn, Hasan H. Otu, Long Ngo, Tamara Fong, Xuesong Gu, Michele Cavallari, Alexandra Touroutoglou, Mouhsin Shafi, Sharon K. Inouye, Zhongcong Xie, Edward R. Marcantonio, Towia A. Libermann

**Affiliations:** 1Division of Interdisciplinary Medicine and Biotechnology, Beth Israel Deaconess Medical Center, Boston, MA 02215, USA; sdillon1@bidmc.harvard.edu (S.T.D.); xgu@bidmc.harvard.edu (X.G.); 2Beth Israel Deaconess Medical Center Genomics, Proteomics, Bioinformatics and Systems Biology Center, Boston, MA 02215, USA; 3Harvard Medical School, Boston, MA 02215, USA; svasunil@bidmc.harvard.edu (S.M.V.); lngo@bidmc.harvard.edu (L.N.); tfong@bidmc.harvard.edu (T.F.); miches@bwh.harvard.edu (M.C.); atouroutoglou@mgh.harvard.edu (A.T.); mshafi@bidmc.harvard.edu (M.S.); zxie@mgh.harvard.edu (Z.X.); emarcant@bidmc.harvard.edu (E.R.M.); 4Divisions of General Medicine, Department of Medicine, Beth Israel Deaconess Medical Center, Boston, MA 02215, USA; 5Departments of Epidemiology and Biostatistics, Harvard T.H. Chan School of Public Health, Boston, MA 02115, USA; 6Department of Electrical and Computer Engineering, University of Nebraska-Lincoln, Lincoln, NE 68588, USA; hotu2@unl.edu; 7Department of Neurology, Beth Israel Deaconess Medical Center, Boston, MA 02215, USA; 8Aging Brain Center, Marcus Institute for Aging Research, Hebrew Senior Life, Boston, MA 02131, USA; sharoninouye@hsl.harvard.edu; 9Center for Neurological Imaging, Department of Radiology, Brigham and Women’s Hospital, Boston, MA 02115, USA; 10Frontotemporal Disorders Unit, Department of Neurology, Massachusetts General Hospital, Boston, MA 02114, USA; 11Berenson-Allen Center for Noninvasive Brain Stimulation, Beth Israel Deaconess Medical Center, Boston, MA 02215, USA; 12Divisions of Gerontology, Department of Medicine, Beth Israel Deaconess Medical Center, Boston, MA 02215, USA; 13Department of Anesthesia, Critical Care and Pain Medicine, Massachusetts General Hospital, Boston, MA 02114, USA

**Keywords:** cerebrospinal fluid, pathophysiology, postoperative delirium, protein biomarkers, proteomics, SomaScan

## Abstract

Delirium is a common postoperative complication among older patients with many adverse outcomes. Due to a lack of validated biomarkers, prediction and monitoring of delirium by biological testing is not currently feasible. Circulating proteins in cerebrospinal fluid (CSF) may reflect biological processes causing delirium. Our goal was to discover and investigate candidate protein biomarkers in preoperative CSF that were associated with the development of postoperative delirium in older surgical patients. We employed a nested case–control study design coupled with high multiplex affinity proteomics analysis to measure 1305 proteins in preoperative CSF. Twenty-four matched delirium cases and non-delirium controls were selected from the Healthier Postoperative Recovery (HiPOR) cohort, and the associations between preoperative protein levels and postoperative delirium were assessed using *t*-test statistics with further analysis by systems biology to elucidate delirium pathophysiology. Proteomics analysis identified 32 proteins in preoperative CSF that significantly associate with delirium (*t*-test *p* < 0.05). Due to the limited sample size, these proteins did not remain significant by multiple hypothesis testing using the Benjamini–Hochberg correction and q-value method. Three algorithms were applied to separate delirium cases from non-delirium controls. Hierarchical clustering classified 40/48 case–control samples correctly, and principal components analysis separated 43/48. The receiver operating characteristic curve yielded an area under the curve [95% confidence interval] of 0.91 [0.80–0.97]. Systems biology analysis identified several key pathways associated with risk of delirium: inflammation, immune cell migration, apoptosis, angiogenesis, synaptic depression and neuronal cell death. Proteomics analysis of preoperative CSF identified 32 proteins that might discriminate individuals who subsequently develop postoperative delirium from matched control samples. These proteins are potential candidate biomarkers for delirium and may play a role in its pathophysiology.

## 1. Introduction

Postoperative delirium is a significant and serious complication after anesthesia and surgery in older adults, occurring in 15–50% of all major surgeries [1]. Delirium often has profound short- and long-term functional and cognitive impacts, and is associated with increased postoperative complications, longer hospital stays, greater discharge rates to nursing homes, and higher in-hospital mortality [2]. Recent evidence suggests that delirium may increase the risk of long-term cognitive decline, with the potential for subsequent development of Alzheimer’s disease and related dementias (ADRD) [3,4].

Identifying postoperative delirium biomarkers has assumed increased importance for risk prediction and advancing pathophysiologic understanding. This cognitive disorder could become a predictable condition if the pathophysiological processes that contribute to delirium risk were better characterized at the molecular and cellular levels. One current model of delirium describes a pre-inflammatory state leading to the development of delirium that is triggered by a precipitating event such as surgery in individuals with a vulnerable brain, resulting in the development of delirium [5]. The vulnerable brain typically indicates underlying or undetected neural dysfunction [2,6], and the risk of vulnerability increases with advancing age. In peripheral blood of surgical patients, our prior work has demonstrated that delirium is characterized by higher preoperative levels of the inflammatory markers C-reactive protein (CRP) and chitinase 3-like protein-1 (CHI3L1/YKL-40), as well as a postoperative hyperactive inflammatory response [7,8,9]. Peripheral inflammation induced by surgery can result in a more permeable blood–brain barrier (BBB), which triggers neuroinflammation, neuronal damage, and consequently delirium in vulnerable individuals [10].

Cerebrospinal fluid (CSF) is the most proximal fluid to the biological processes in the brain and is a biofluid accessible during induction of spinal anesthesia that may closely reflect the state of the brain preoperatively. Targeted biomarker studies in preoperative CSF have identified several cytokines, one chemokine and cholinergic biomarkers associated with the development of postoperative delirium [11,12,13]. One untargeted study using mass spectrometry-based proteomics on preoperative CSF in patients with hip fracture identified 63 protein changes that were associated with postoperative delirium [14]. We report a comparison of 1305 proteins analyzed by high multiplex, aptamer-based affinity SomaScan proteomics analysis from the preoperative CSF of older patients undergoing elective orthopedic surgery under spinal anesthesia. SomaScan technology was selected due to its broad dynamic range spanning more than 10 logs of protein concentration and the assay’s ability to measure every target protein in each sample with no missing values [15]. The study design included 24 matched-pair samples from patients diagnosed with delirium matched to patients who did not develop delirium from the Healthier Postoperative Recovery (HiPOR) study cohort. The primary objective was to conduct an exploratory, hypothesis-generating study by applying proteomics analysis to preoperative CSF samples to investigate preoperative proteins with expression changes associated with delirium risk in response to major surgery and to use these associations to probe specific pathophysiologic pathways.

## 2. Materials and Methods

### 2.1. Study Design and Human Subject Population

This was a retrospective nested case–control investigation using the Healthier Postoperative Recovery (HiPOR) sample cohort. Overall study design is presented in Figure 1. The HiPOR study protocol was approved by the Partners Human Research Committee (Boston, MA, USA), and all participants provided written informed consent. Eligible individuals were aged 63 years or older and were admitted for elective total knee and hip replacement using spinal anesthesia at the Massachusetts General Hospital [16,17]. Older adults with prior dementia were excluded based on patient or family report of dementia diagnosis, medical record review, or a baseline Mini-Mental State (MMSE) [18] score of less than 24. For the current study, only participants enrolled between 2009 and 2016 with adequate banked CSF specimens were used.

### 2.2. Assessment of Delirium and Severity in HiPOR

The overall delirium rate was 20% in the participants enrolled in HiPOR. Delirium was assessed using a structured cognitive battery administered on postoperative days (POD) 1 and 2. The battery consisted of cognitive testing using the Mini-Mental State Examination (MMSE), purchased from Psychological Assessment Resources (PAR), Inc., and patient symptom reporting based on the Delirium Symptom Interview (DSI) [19] to score the Confusion Assessment Method (CAM). Delirium severity measures were not available for HiPOR. Therefore, days of delirium was employed as a proxy for severity. See Appendix A for more details concerning the scoring of delirium and severity.

### 2.3. Collection of Patient CSF

All HiPOR participants underwent spinal anesthesia for surgery. During induction of spinal anesthesia, 1 mL of CSF was obtained from a spinal needle prior to the administration of local anesthetic. CSF was collected via dropwise collection into collection tubes. To minimize potential blood contamination of the CSF, samples were centrifuged at 1000× *g* for 10 min prior to subaliquoting and storage at −80 °C.

### 2.4. Selection of Nested Matched Case–Control Samples for Proteomics Analysis

To maximize power for biomarker discovery on a small sample size, we created a nested, matched case–control set of 24 pairs from the HiPOR cohort. Delirium cases were defined as meeting CAM delirium criteria on POD 1 and 2. Eligible non-delirium controls were patients without delirium or subsyndromal delirium (SSD) on any postoperative day. Patients with SSD were ineligible to be considered as cases or controls. Cases and controls were matched using the optimal method [20] based on four patient criteria: gender, age (within 5 years), year of surgery (within 2 years), and preoperative MMSE score (within 3 points).

### 2.5. Proteomics Analysis Using SomaScan

CSF samples from all 24 matched case–control pairs, 48 samples in total, were analyzed by the SomaScan proteomics assay from SomaLogic (Boulder, CO, USA) at the BIDMC Genomics, Proteomics, Bioinformatics and Systems Biology Center according to the standard protocol for CSF developed by SomaLogic. Additional details describing the SomaScan assay are located in the Appendix A. The full SomaScan dataset can be found as a Appendix A. The data have been submitted to GEO with the accession number GSE242736.

### 2.6. Statistical Analysis

#### 2.6.1. Identification of a Preoperative CSF Proteomic Signature of Postoperative Delirium

Differential expression analysis compared delirium cases and non-delirium control samples preoperatively using paired *t*-tests. A protein was differentially expressed if it was significantly different in cases and controls (*p* < 0.05). Multiple hypothesis testing correction was performed using both the Benjamini–Hochberg (BH)-correction [21] and q-value calculation [22]; however, neither method found significant proteins after correction. Therefore, raw *p*-values were used to define the differentially expressed protein list. Fold-change (FC) of protein expression was calculated for each pair of samples (delirium/non-delirium control) and downregulation in the delirium group was indicated by a negative sign. For example, an FC of 2 would imply a 2-fold upregulation in the delirium sample while an FC of -2 would imply a 2-fold downregulation in the delirium sample for the pair. The FC value for a protein was calculated by applying the one-step Tukey’s biweight algorithm to the FC values for each paired sample to minimize outlier effects (tFC) [23]. Tukey’s algorithm provides a robust estimate for the protein FC values, with positive and negative values indicating up- and downregulation in the delirium group, respectively. Participant samples and proteins were clustered using the unweighted pair group method with arithmetic mean (UPGMA), also known as hierarchical clustering (HCA), with Pearson’s correlation as the similarity measure [24]. For HCA, each protein expression vector was standardized to have zero mean, unit variance across the samples, and average linkage was employed in the agglomeration step of the UPGMA algorithm.

Sample classification was applied to two-dimensional principal component analysis (2DPCA) transformed values using support vector machines (SVMs) running linear kernels [25]. To avoid overfitting, classification accuracy was assessed using leave-one-out cross-validation. Classification utility of the protein signature independent of model cut-off parameters was measured by applying the area under the curve (AUC) of the receiver operating characteristic (ROC) curve analysis [26]. Machine learning analyses were performed in MATLAB (v.2021b, The MathWorks Inc., Natick, MA, USA). Statistical tests were performed either in MATLAB or the XLSTAT (Addinsoft, New York, NY, USA, 2022) add-on for Excel.

#### 2.6.2. Molecular and Systems Biology Analysis

The data analytic methods employed in this study are summarized in Figure 1. Three complementary analysis methods were utilized to gain insights into the molecular functions and pathways of identified proteins. Gene Ontology (GO) to elucidate molecular and biological functions was performed on all 32 proteins leveraging the Protein Knowledgebase component of the UniProt database (https://www.uniprot.org/uniprot/, accessed on 20 January 2022) [27]. Pathophysiological pathways underlying delirium-specific preoperative CSF protein signatures were determined by the Ingenuity Pathway Analysis (IPA) software tool version 01-22-01 (QIAGEN, Redwood City, CA, USA) [28]. Additional details are presented in the Appendix A.

## 3. Results 

### 3.1. Nested Case–Control Sample Characteristics

This was a retrospective nested case–control investigation using the HiPOR cohort. Figure 1 presents the overall study design. From the HiPOR study (n = 289), 24 delirium cases were matched with 24 non-delirium controls based on four patient criteria (gender, age, year of surgery, and preoperative MMSE score). Mean age for delirium patients was 73.0, and for the non-delirium controls, it was 72.6 (Table 1). The percentage of female patients was 46% in both cases and controls. MMSE scores were 27.2 for delirium patients and 27.5 for non-delirium controls. Though not a match variable, level of education was similar in cases and controls. The type and overall distribution of co-morbidities identified from the medical records for all 48 patients showed no significant differences between delirium cases and non-delirium controls (Appendix A).

### 3.2. Proteomics Analysis Identifies Preoperative Proteins in CSF Associated with Postoperative Delirium

Proteomics analysis of 1305 proteins was performed on preoperative CSF samples obtained from all 48 individuals in the 24 matched pairs. Applying the paired *t*-test, 32 proteins were identified to be significantly different between the participants with development of postoperative delirium and the participants without development of postoperative delirium (*p*-value < 0.05). Eighteen proteins were elevated, while 14 proteins were decreased, in delirium cases (Table 2).

The ability of the 32 proteins to discriminate between delirium and non-delirium in the 48 analyzed samples was assessed by three separate analyses of the protein expression data. Hierarchical cluster analysis (HCA), an unsupervised learning method, demonstrated these 32 proteins were different between delirium cases and non-delirium controls with a classification success rate of 83.3% (40/48) (Figure 2A). Principal components analysis (PCA) combined with machine learning (SVM) resulted in separation of the delirium cases from the controls into two clusters with a leave-one-out cross-validation accuracy of 89.6% (Figure 2B). Finally, a SVM model based on all 32 proteins generated a receiver operating characteristic (ROC) curve with an area under the curve (AUC) of 0.91 [0.80–0.97] (95% confidence interval is denoted in brackets) (Figure 2C).

### 3.3. Systems Biology Analysis of the 32 CSF Proteins Discriminating Delirium from Non-Delirium Cohorts

Several common functional groups were revealed after Gene Ontology (GO) analysis of the 32 proteins. Cytokines, chemokines, growth factors and hormones are secreted proteins that act on their local environment as well as on distant cells to regulate organismal function. We identified seven of these proteins (C-X-C motif chemokine 11 (CXCL11), C-X-C motif chemokine 6 (CXCL6), C-C motif chemokine 2 (CCL2), C-C motif chemokine 28 (CCL28), thrombopoietin (THPO), insulin (Ins) and β-nerve growth factor (NGF)) that represent the largest functional group (7/32 = 22%). Several of the altered proteins interact with the actin cytoskeleton: cofilin-1 (CFL1), moesin (MSN), matrix metalloproteinase-14 (MMP-14) and CCL2. The nicotinamide adenine dinucleotide (NAD+) biosynthetic pathway was represented by two functional enzymes (nicotinamide phosphoribosyltransferase (NAMPT) and ADP-ribosyl cyclase/cyclic ADP-ribose hydrolase 1 (CD38)). Three proteases (cathepsin L2 (CTSD), cathepsin D (CTSV) and MMP-14) were also among the 32 significant proteins.

Biological functional analyses using Ingenuity Pathway Analysis (IPA) revealed 30 pathways with significant enrichment for the 32 proteins (Figure 3). “Inflammatory response” was the most significant pathway and “inflammation of organ” was also strongly associated. The majority of identified pathways involved “immune cell movement, function and turnover” (24/30 = 80%). Fifty percent (16/32) of the CSF altered proteins demonstrated an association with the inflammatory response (Appendix A). The accumulated effect (activation z-score = −0.741) from the differential proteins was predicted to reduce the inflammatory response in CSF in delirium patients prior to surgery. Migration and chemotaxis of immune cells exhibited significant associations with negative z-scores ranging from −1.54 to −2.353, indicating reduced movement of myeloid cells (Appendix A). Sixty-nine percent (22/32) of the proteins were related to apoptosis, and were predicted to inhibit this pathway prior to onset of delirium (Appendix A). Two pathways with connection to central nervous system (CNS) function were predicted to be activated: long-term synaptic depression (6/32 = 19%) (z-score = 0.24) (Appendix A) and neuronal cell death (10/32 = 31%) (z-score = 1.693) (Appendix A). Finally, an inhibitory effect on angiogenesis was predicted (12/32 = 38%) (z-score = −1.981) (Appendix A).

Further analysis to identify shared upstream regulatory proteins for the 32 CSF proteins revealed 20 significant regulators associated with the biological functions described above, the majority encoding cytokines and chemokines (Appendix A). Upstream regulator models predicted the cytokines tumor necrosis factor-α (ΤΝFα) (z-score = 1.493) (Appendix A), interferon-γ (ΙFNγ) (z-score = 0.848) (Appendix A), interleukin-6 (IL6) (z-score = 0.823) (Appendix A) and interleukin-10 (IL10) (z-score = 1.414) (Appendix A) to be potentially activated and transforming growth factor-β1 (TGF-β1) (z-score = −1.057) (Appendix A) to be inhibited. Two chemokines were identified as significant upstream regulators: CC-motif chemokine 11 (CCL11) and CC-motif chemokine 5 (CCL5). Among the pro-inflammatory cytokines, TNFα is upstream of 17 of the 32 CSF delirium-associated proteins (Appendix A), IFNγ has a potential connection to 15/32 CSF proteins (Appendix A), IL-6 is upstream of 10/32 proteins (Appendix A), and IL-10 (Appendix A) is upstream of 8/32 proteins. In contrast, TGFβ1 may negatively impact 5 of 13 predicted downstream proteins (Appendix A).

Next, we performed network and cluster analysis using STRING. Twenty-six of the 32 proteins formed distinct interacting protein clusters enriched in pathways associated with chemokine signaling, extracellular matrix organization and degradation, oxidative stress and insulin-like growth factor (IGF) stimulated transport (Figure 4). Insulin occupies a key focus hub in this analysis, linking extracellular matrix organization and degradation, chemokine signaling and oxidative stress. Our proteomics analysis revealed that insulin may be elevated before surgery in the CSF of patients who develop delirium.

### 3.4. Analysis of Delirium Severity Correlates Significantly with Many of the Preoperative Proteins Associated with Postoperative Delirium

Spearman’s correlation analysis of the proteomics data of delirium severity identified 23 proteins with significance (*p* < 0.05) (Appendix A). A smaller number of proteins was expected, as the unmatched design reduced statistical efficiency. Ten of these 23 proteins were increased, and 13 proteins decreased. The *t*-test *p*-value based on incidence of postoperative delirium for each of these 23 proteins is presented. Overall, 14 proteins (indicated by ** in Appendix A) were in common and significant (*p* < 0.05) between the 24 matched samples analyzed for postoperative delirium incidence versus the 48 unmatched samples for delirium severity (Appendix A). The combined list of unique proteins significant for either incidence or severity totaled 41 proteins (Appendix A). Applying a slightly less stringent *p*-value cutoff of 0.1 to either incidence (indicated by *) or severity (indicated by #) identified an additional 17 proteins. Thus, 31 out of the combined total of 41 unique proteins (76%) were supported by significance or near significance for the analysis of the matched delirium incidence and the unmatched delirium severity (Appendix A).

## 4. Discussion

Employing SomaScan proteomics analysis, we investigated the proteome of preoperative CSF in 48 matched case–control samples from older patients undergoing elective joint replacement surgery to identify potential protein biomarkers associated with the risk of postoperative delirium. We identified 32 CSF proteins that differed significantly before surgery between the participants who developed versus did not develop postoperative delirium. However, these proteins did not remain significantly dysregulated by multiple hypothesis testing after BH correction or q-value calculation. Nevertheless, our successful clustering and classification results provided strong associations of these 32 proteins with delirium status, and our functional analysis at the systems level showed high clinical and biological relevance. Three algorithms (HCA, PCA and ROC) were applied to assess the ability of the 32 proteins to discriminate delirium from non-delirium. HCA correctly clustered 40/48 (83.3%) of delirium cases versus non-delirium controls. PCA was able to separate 43/48 (89.6%) of the analyzed samples. ROC curve analysis generated an AUC of 0.91. Although expression patterns shown in HCA exhibited subtle differences between delirium and non-delirium samples for individual proteins, collectively, the 32-protein signature demonstrated a strong association with delirium based on our clustering and classification results. Protein functional and pathway analyses highlighted several key biological processes that may reveal delirium pathophysiology. These included inflammation, migration of immune cells, cell death, neuronal injury, and angiogenesis. Our results indicate that pre-surgical alterations may play a role in the susceptibility for the development of delirium and may represent some features of a vulnerable brain.

This is the first study using an innovative, globally targeted proteomics approach employing the aptamer-based SomaScan platform to measure preoperative protein differences in the CSF from patients that did, and did not, develop postoperative delirium. SomaScan proteomics analysis has been successfully applied in a few studies to analyze human and rodent CSF [29,30]. For delirium, the CSF proteome has been measured in published reports employing mass spectrometry (MS)-based protein identification. Common proteins among all three MS-based studies did not emerge [14,31,32]. Moreover, none of the MS-identified delirium-altered proteins overlapped with the 32 proteins identified in this study. Direct comparison of results across unique patient populations, applying different protein separation techniques and instrumentation for global protein analysis, is challenging to identify shared statistically significant protein changes [33]. However, additional cross-study confirmation can be derived from common altered biological pathways [34]. Poljak et al., 2014 [31] and Westoff et al., 2015 [32] both detected proteins altered in the acute-phase response, which is part of the inflammatory pathway, in agreement with our study’s identification of the inflammatory response. Han et al., 2020 [14] identified dysregulation of neural pathways, as well as insulin-like growth factor (IGF) and its binding proteins (IGFBPs). Our study also found neural related pathways (long-term synaptic depression and neuronal cell death) and insulin and IGFBP-2 altered in CSF in support of the pathway results from Han et al., 2020 [14]. Most importantly, the lack of specific overlapping proteins across the different studies can also be explained as a consequence of striking differences between the patient cohorts and time of CSF collection. For example, Poljak et al., 2014 [31] studied a cohort of patients with delirium triggered by infections, metabolic problems and adverse drug reactions at the time of CSF collection, in contrast to our study of preoperative CSF before any onset or trigger of delirium. Han et al., 2020 [14] and Westoff et al., 2015 [32] measured proteins in preoperative CSF from patients with an acute hip fracture injury, which may already trigger inflammation and distress as compared to our elective orthopedic surgery.

We observed the inflammatory pathway as the most highly enriched pathway in CSF for our 32 proteins associated with delirium risk. This enrichment was primarily driven by proteins associated with migration and chemotaxis of the myeloid lineage. Reduced expression of several chemokines predict a decrease in movement of myeloid cells. In contrast, upstream regulator analysis suggests increased activity of pro-inflammatory cytokines such as TNF, IFNγ, and IL6, which would be expected to increase inflammatory processes. While these results appear counterintuitive, the inflammatory response is complex, and different branches of this response may act independently and with unique directionality. Impaired regulation of the inflammatory response due to advancing age and a low-level rise in baseline inflammation (so-called inflammatory priming) are proposed as leading to an inappropriately regulated overstimulation due to surgery [9]. The predicted enhanced activity of proinflammatory cytokines on proteins in the CSF may support this notion. Our results demonstrated several inflammatory mediators specifically altered in the preoperative CSF of delirium patients. CXCL1, increased in the CSF of delirium cases, has been previously shown to be elevated in the CSF of people with neuroinflammation [35]. In agreement with the decreased expression in our results, CXCL6 is decreased in the CSF of patients experiencing primary progressive aphasia, a sporadic neurodegenerative disease in the frontotemporal dementia spectrum associated with neuroinflammation [36]. CCL2 (also known as MCP-1) is elevated immediately after surgery in patients with delirium [37]. CCL28 has been measured to be decreased in tissue sections from Parkinson’s disease (PD) patients [38]. THPO, an activating cytokine, has been proposed to act early in dementia, and insulin-like growth factor binding protein-2 (IGFBP2) plays a modifying role [39].

Since delirium affects brain function, we predict that some proteins in our CSF discriminatory panel should demonstrate a direct link to neural function. Two proteins in our panel, NGF and Parkinson disease protein 7 (PARK7), have well-established connections to brain function and dementia. Our proteomics analysis measures NGF as increased in patients that will develop delirium, and it has been shown that NGF is increased in the CSF of patients with AD [40]. We found that PARK7 is increased in patients that will develop delirium, and mutations in the PARK7 gene are directly involved in early-onset PD [41]. An increase in PARK7 in the CSF of patients that will develop delirium suggests inappropriate levels of oxidation may be present. The proper turnover and removal of proteins by controlled homeostatic proteolysis demonstrates a key role in developing and maintaining a healthy brain [42]. We observe decreased levels in three proteases (CTSV, CTSD and MMP14) in the CSF of patients that will develop delirium.

Disruptions in brain energy metabolism have been connected to delirium [43]. We detect elevated insulin in CSF prior to surgery-induced delirium, and an altered carbohydrate metabolism is involved in a brain vulnerable to cognitive disorders [44]. An established connection exists between glucose metabolism, cognition and neurodegenerative diseases such as AD [45]. An association with delirium is beginning to emerge as well and may indicate a contributory effect of energy metabolism disruption [46]. Insulin receptors are widely distributed in the brain with concentration in areas involved in cognition and regulation of food intake [45]. The NAD+ pathway is critical to maintaining healthy brain homeostasis [47]. NAMPT and CD38, two enzymes catalyzing direct steps in the production and consumption of NAD+, are both elevated in CSF prior to surgery in delirium patients. Disturbances in the levels of enzymes that control NAD+ levels in the brain result in aberrant neurological function and are found in several neurological diseases (AD, PD, amyotrophic lateral sclerosis and Lewy body dementia).

## 5. Strengths and Limitations

The major strengths of this study are the use of prospectively collected and well-characterized preoperative CSF samples from a cohort of older adults undergoing major orthopedic surgery. Proteomics analysis enabled us to query 1305 proteins using a sensitive and global proteomics technique. Many of the proteins among our 32-protein discriminatory CSF panel demonstrate direct or plausible associations with cognitive and neural disorders and may reveal underlying pathophysiology. Biological pathway analysis demonstrated that many of the 32 proteins are biologically connected in specific pathways, indicating that the set of proteins exhibits plausible biology and is unlikely to be random. This analysis illustrates a key role for inflammation and neural-related pathways. Delirium severity analysis found 14 significant proteins that are shared with the 32 proteins identified for postoperative delirium.

Our study had important limitations. It was limited by the small sample size not providing enough power to identify proteins that remain significant after multiple hypothesis testing correction. However, our clustering and classification results showed successful discriminatory power of our 32 proteins for delirium, and our systems biology analysis demonstrated strong biological and clinical relevance. Sample size was limited by the matched-pairs selection criteria and the volume limitations of the HiPOR samples (many of which were used for previous studies and not available) and thus precluded examination of delirium subtypes other than severity. The small sample size did not allow us to split our protein data into a training and validation set. Thus, the validity of our model remains to be further tested using additional CSF samples. Finally, blood and postoperative CSF were not collected, meaning that correlative and subsequent changes could not be examined. These are important areas for future work. As our study is exploratory, our analysis should be regarded as hypothesis-generating. Future work will be required to confirm and validate the present findings in independent samples.

## 6. Conclusions

In this first study analyzing the CSF proteome of delirium using SomaScan analysis, we report a preoperative protein signature in the CSF of surgical patients with postoperative delirium that includes 32 proteins. Eighteen proteins show an increase in CSF, while fourteen proteins are decreased. The identified proteins participate in key physiological functions: inflammation, immune cell migration, apoptosis, angiogenesis, synaptic depression and neuronal cell death, providing new insights into the pathophysiology of delirium.

## Figures and Tables

**Figure 1 biomolecules-13-01395-f001:**
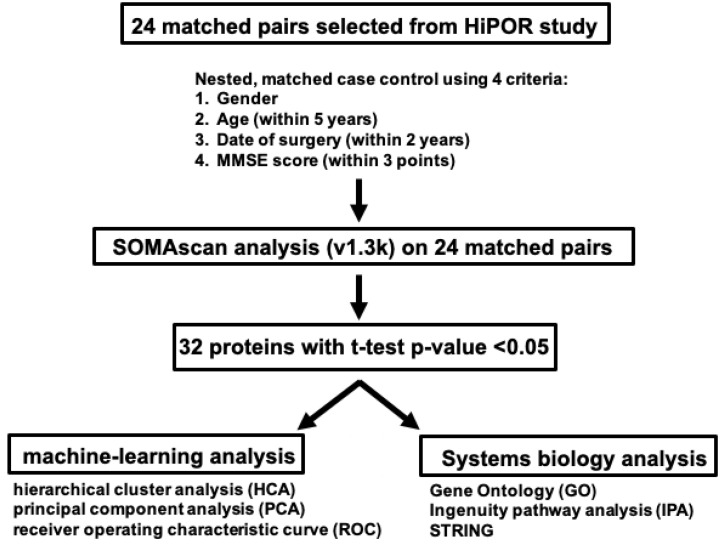
CSF Proteomics Analysis Study Design on HiPOR Samples. Diagram depicting the workflow in the proteomics analysis of HiPOR CSF samples. Twenty-four matched pairs (48 samples) were analyzed by SomaScan v1.3k. Thirty-two proteins were identified with a *t*-test *p*-value < 0.05. Unsupervised machine learning techniques and systems biology analyses were applied to the 32 significant proteins.

**Figure 2 biomolecules-13-01395-f002:**
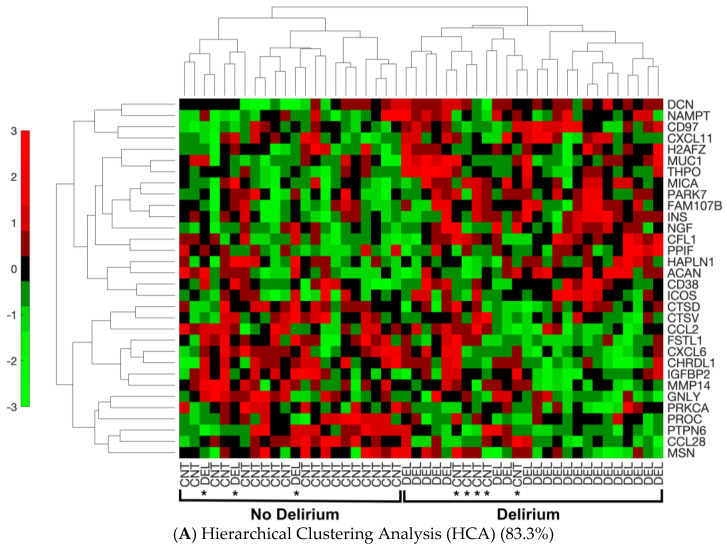
Separation of Delirium Versus Non-Delirium Using the 32 Significant Differential Proteins by Three Unique Linear Analysis Methods. (**A**). Heat map derived from the machine-learning Hierarchical Cluster Analysis (HCA) algorithm reflecting relative minimum and maximum expression levels for each given protein among patients as determined by proteomics analysis results. Comparisons were made between all patients with delirium and non-delirium control subjects. *p* < 0.05 comparing delirium to matched non-delirium controls for the 32 differential proteins is shown. In the HCA heat map, the color bar represents the display range, [−3, 3]; for the standardized signal values mapped to the red–green continuum, red denotes upregulation and green denotes downregulation. Protein names are represented by the gene symbol, and full names are presented in Table 2. CNT = control non-delirium samples; DEL = delirium samples. * indicates an incorrect classification: this means a CNT sample clusters with the DEL samples or vice versa. (**B**). Principal components analysis (PCA) separation of patients with delirium versus non-delirium control subjects into two dimensions with support vector machines (SVMs). The separating line is based on a linear kernel using the 32 protein CSF panel derived from proteomics analysis. The first principal component accounts for 22.48% of the variance, and the second principal component for 12.69% of the variance. PCA demonstrates that the derived proteomics data contain a significant component separating delirium cases from matched control samples (43/48 = 89.6%). Red circles denote delirium samples. Blue triangles represent non-delirium controls. (**C**). Receiver operating characteristic (ROC) curve analysis based on the 32 protein delirium signature. Our CSF delirium signature discriminates delirium subjects from non-delirium control subjects with an area under the curve (AUC) = 0.91 [0.80–0.97]. The 95% confidence interval for the AUC is shown in brackets.

**Figure 3 biomolecules-13-01395-f003:**
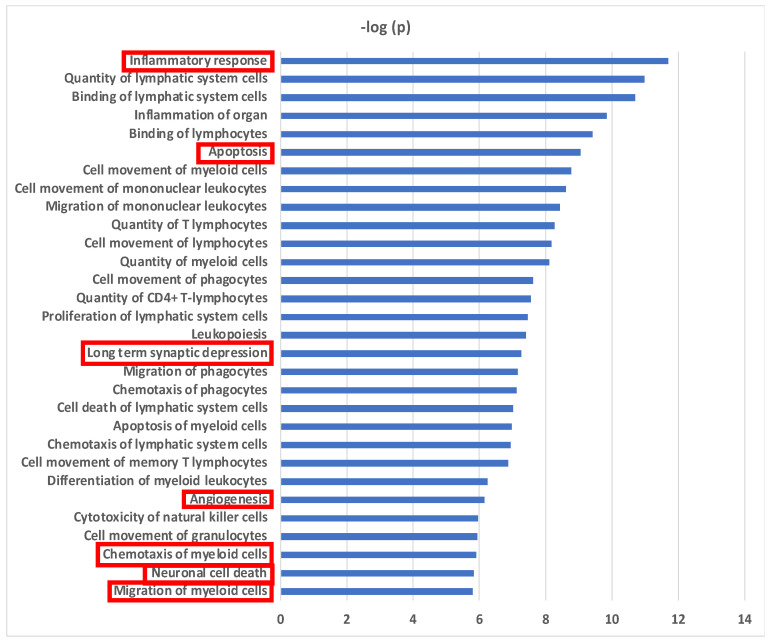
Biological Pathways for Preoperative CSF Proteins in Delirium. Biological functional pathways determined using Ingenuity Pathway Analysis (IPA) that are significantly enriched in the 32 CSF delirium-associated proteins. Level of significance: *p* < 10^−5^. Migration and chemotaxis of immune cells includes these pathways: “Cell movement of monocytes”, “Cell movement of granulocytes”, “Cell movement of myeloid cells”, “Cell movement of phagocytes”, “Quantity of granulocytes”, “Migration of phagocytes”, “Chemotaxis of granulocytes”, “Chemotaxis of myeloid cells”, “Migration of myeloid cells”, and “Migration of monocytes”. Pathways enclosed by a red rectangle are presented in Appendix A.

**Figure 4 biomolecules-13-01395-f004:**
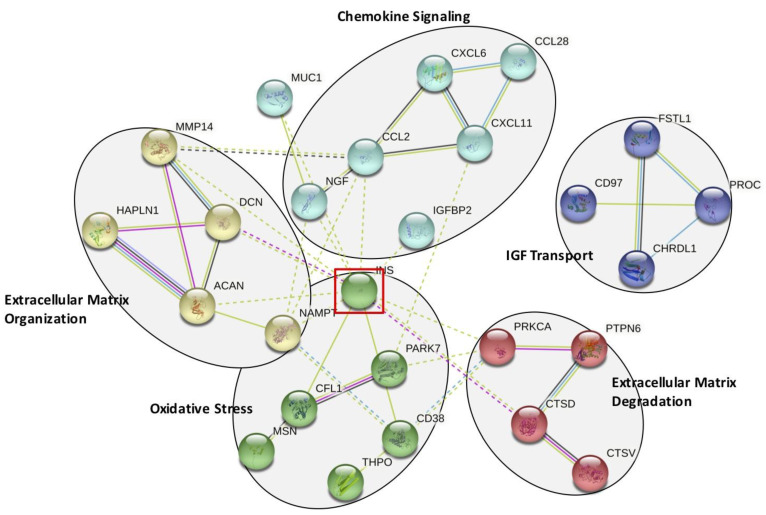
STRING Analysis of the 32 Significantly Altered CSF Proteins. Functional pathways identified using the STRING analysis program. Five functional groups are identified: chemokine signaling, extracellular matrix organization and degradation, oxidative stress, and insulin-like growth factor (IGF) mediated transport. This analysis identifies insulin (INS) as a key interacting protein, highlighted by the red-lined box.

**Table 1 biomolecules-13-01395-t001:** Baseline Characteristics of HiPOR Delirium Cases and Non-Delirium Controls.

	HiPOR 24 Matched Pairs (n = 48)	
Characteristic	Delirium Cases (n = 24)	Non-Delirium Controls (n = 24)
Age (M, SD)	73.0 (4.9)	72.6 (5.5)
Female, n (%)	11 (46)	11 (46)
MMSE (M, SD)	27.2 (2.0)	27.5 (1.7)
Year of surgery, n (%)		
2009–2010	5 (21)	2 (8)
2011–2012	8 (33)	13 (54)
2013–2014	6 (25)	6 (25)
2015–2016	5 (21)	3 (13)
Highest level of education, n (%) *		
<12 years	0 (0)	1 (7)
12 years or GED	1 (8)	1 (7)
Some college **	1 (8)	5 (33)
College graduate or postgraduate school	11 (84)	8 (53)
Days of delirium, n (%)		
no delirium	0 (0)	24 (100)
1 day (mild)	20 (83)	0 (0)
2 days (severe)	4 (17)	0 (0)

M = mean, SD = standard deviation, MMSE = Mini-Mental State Examination. * Data available for n = 28 (Delirium cases n = 13, Non-delirium controls n = 15). ** Includes vocational/technical program.

**Table 2 biomolecules-13-01395-t002:** Upregulated and Downregulated Proteins Between Matched Delirium and Non-Delirium Samples.

Increased in Delirium				
Full Name of Protein	UniProt	Gene Symbol	tFC	*p*-Value
Aggrecan core protein	P16112	ACAN	1.79	0.013
Cofilin-1	P23528	CFL1	1.62	0.041
C-X-C motif chemokine 11	O14625	CXCL11	1.46	0.047
Histone H2A.z	P0C0S5	H2AFZ	1.44	0.020
Mucin-1	P15941	MUC1	1.39	0.039
Nicotinamide phosphoribosyltransferase	P43490	NAMPT	1.36	0.029
Insulin	P01308	INS	1.35	0.019
CD97 antigen	P48960	CD97	1.32	0.006
Inducible T-cell costimulator	Q9Y6W8	ICOS	1.31	0.028
Protein deglycase DJ-1	Q99497	PARK7	1.31	0.030
Protein FAM107B	Q9H098	FAM107B	1.29	0.028
ADP-ribosyl cyclase/cyclic ADP-ribose hydrolase 1	P28907	CD38	1.29	0.048
Beta-nerve growth factor	P01138	NGF	1.2	0.020
Peptidyl-prolyl cis-trans isomerase F, mitochondrial	P30405	PPIF	1.17	0.035
Thrombopoietin	P40225	THPO	1.12	0.010
Decorin	P07585	DCN	1.09	0.013
MHC class I polypeptide-related sequence A	Q29983	MICA	0.9	0.011
Hyaluronan and proteoglycan link protein 1	P10915	HAPLN1	0.86	0.042
**Decreased in Delirium**				
Cathepsin L2	O60911	CTSV	−1.47	0.048
Granulysin	P22749	GNLY	−1.43	0.045
C-X-C motif chemokine 6	P80162	CXCL6	−1.36	0.017
C-C motif chemokine 2	P13500	CCL2	−1.34	0.030
Matrix metalloproteinase-14	P50281	MMP14	−1.29	0.030
Moesin	P26038	MSN	−1.24	0.014
Chordin-like protein 1	Q9BU40	CHRDL1	−1.22	0.035
Activated Protein C	P04070	PROC	−1.21	0.015
C-C motif chemokine 28	Q9NRJ3	CCL28	−1.21	0.015
Cathepsin D	P07339	CTSD	−1.15	0.021
Follistatin-related protein 1	Q12841	FSTL1	−1.12	0.040
Insulin-like growth factor-binding protein 2	P18065	IGFBP2	−1.07	0.028
Tyrosine-protein phosphatase non-receptor type 6	P29350	PTPN6	−1.04	0.044
Protein kinase C alpha type	P17252	PRKCA	−1.02	0.035

tFC = Tukey Fold Change, *t*-test *p*-value < 0.05 is considered significant.

## Data Availability

The datasets used and/or analyzed during the current study are available from the corresponding author on reasonable request and are available in the Appendix A and under GEO accession number GSE242736.

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
