# Peer review of "Aptamer-Based Proteomics Measuring Preoperative Cerebrospinal Fluid Protein Alterations Associated with Postoperative Delirium"

_biomolecules, 2023, doi:10.3390/biom13091395_

Round 1
Reviewer 1 Report
The manuscript entitled "Aptamer-Based Proteomics Measuring Preoperative Cerebrospinal Fluid Protein Alterations Associated with Postoperative Delirium", by Dillon and coworkers, is an interesting work that aims to identify putative protein biomarkers of the delirium affecting several old patients after surgery, and to characterize the biological pathways related to these proteins. Employing a targeted proteomic approach for the analysis of the cerebrospinal fluid of 24 matched pair samples, the authors found 32 modulated proteins, evaluated their predictivity of delirium and described the related functional pathways.
According to this reviewer the manuscript is well written and interesting, and deserves publication on Biomolecules upon the answer to the following points:
1) The title of the paper introduces the aptamer-based proteomic method, but the space dedicated to this approach in the text is limited. A short description of the method (specifing why the authors chose it instead of an untargeted one) should be added.
2) The Tukey fold change (tFC) values reported in several tables could be difficult to be interpreted for people not familiar with the method: does it correspond to log(2) of the FC? What is the threshold used (usually >1 and <-1, but in Table 3 there are values like 0.9 and 0.86 for "increased in delirium" proteins)?
3) The manuscript is too long and has too many figures. In order to help the reader, I would reorganize the text and move some figures and tables in SI (e.g. Table 2, Figure 19, etc.) and unify several figures related to the same topic (Figures 2-4; Figures 6-11; Figures 13-17).
Reviewer 2 Report
The authors investigated candidate protein biomarkers in preoperative CSF that were associated with development of postoperative delirium in older surgical patients. They employed a nested case–control study design coupled with high multiplex affinity proteomics analysis to measure 1305 proteins in preoperative CSF. There are several problems:
1. The authors need to upload the data onto GEO or other public available databases.
2. Were the 32 proteins with t-test p value<0.05 adjusted for multiple test adjustment?
3. Table 2 did not display correctly.
4. In Figure 2, there was no patterns. Have the authors done normalization?
5. In Figure 3, the tangles and dots were out of the x-axis and y-axis.
6. Form Figure 6 to 17, there are all very similar and redundant. They should be deleted if no key information was shown in them.
7. The manuscript was poorly written. It was too long and too tedious. It needs to be re-written.
The authors investigated candidate protein biomarkers in preoperative CSF that were associated with development of postoperative delirium in older surgical patients. They employed a nested case–control study design coupled with high multiplex affinity proteomics analysis to measure 1305 proteins in preoperative CSF. There are several problems:
1. The authors need to upload the data onto GEO or other public available databases.
2. Were the 32 proteins with t-test p value<0.05 adjusted for multiple test adjustment?
3. Table 2 did not display correctly.
4. In Figure 2, there was no patterns. Have the authors done normalization?
5. In Figure 3, the tangles and dots were out of the x-axis and y-axis.
6. Form Figure 6 to 17, there are all very similar and redundant. They should be deleted if no key information was shown in them.
7. The manuscript was poorly written. It was too long and too tedious. It needs to be re-written.
Round 2
Reviewer 2 Report
The authors have improved the quality of the manuscript.